# Method to Change the Through-Hole Structure to Broaden Grounded Coplanar Waveguide Bandwidth

**DOI:** 10.3390/s23094342

**Published:** 2023-04-27

**Authors:** Jiangmiao Zhu, Zhaotong Wan, Kejia Zhao

**Affiliations:** 1Faculty of Information Technology, Beijing University of Technology, Beijing 100024, China; zhujiangmiao@bjut.edu.cn (J.Z.); wzt18513208066@163.com (Z.W.); 2National Institute of Metrology, Beijing 100029, China

**Keywords:** broadband grounded coplanar waveguide, COMSOL Multiphysics, through-hole array, S-parameters

## Abstract

A grounded coplanar waveguide (GCPW), as a millimeter wave special transmission line, can be used to calibrate broadband oscilloscope probes. A method to change the through-hole structure to widen the GCPW is investigated in this paper. The effect of the through-hole array on the band-width of the GCPW is investigated and verified using COMSOL Multiphysics simulation software. Finally, the S-parameters of the fabricated GCPWs are measured by a vector network analyzer, and the results show that they have an insertion loss > −3 dB and return loss < −10 dB in the frequency range of DC to 60 GHz, which satisfies the design requirements.

## 1. Introduction

A grounded coplanar waveguide (GCPW), as a microwave and above-band microstrip line, has a wide range of applications and is an indispensable component for calibrating oscilloscope probes. Because of the wide bandwidth of the current probe, GCPW can be regarded as a subsystem of the calibration probe. According to the time domain measurement theory, the bandwidth of GCPW must be over 2 to 3 times greater than the bandwidth of the probe [1] so that the GCPW transmission line does not affect the calibration accuracy of the probe. This paper focuses on adjusting the through-hole structure of the GCPW to enhance the bandwidth thereof.

The study of using GCPWs with through-hole arrays as test plates to calibrate probes has received increasing attention, and published studies have reported that the bandwidth of GCPWs typically reaches 50 GHz [2] or 40 GHz [3,4]. The wider the bandwidth, the more stringent the size and structure requirements of the GCPW. Umar changed the signal of a GCPW from a single-ended to a differential input to increase the bandwidth to 60 GHz, which resulted in a complex structure that was not conducive to practical applications [5]. Hui Huang suppressed higher order modes by adding a through-hole structure to the GCPW to achieve bandwidths of up to 50 GHz [6], demonstrating that adding through-holes to the GCPW is another effective way to extend its bandwidth.

Advances in microwave technology have contributed to the development of microwave sensors and the improvement of sensor performance. For example, microwave sensors are microwave dielectric spectroscopy and near-infrared spectroscopy techniques combined to obtain more electromagnetic spectrum data, so the use of microwave sensors can be more effective in identifying different substances without the use of a large number of samples for training [7]. The new sensor combines fluorescence and microwave dielectric spectroscopy techniques into the same physical structure, and the accuracy of honey adulteration identification can be improved by using the combination of two different spectral data. Therefore, this paper is of great interest to investigate the extension of microwave transmission line bandwidth [8].

In this paper, the effects of the distance between the through-hole array and the slot line, the through-hole to through-hole distance, the distance between the GCPW ends and the edge of the through-hole array, different numbers of through-hole arrays, and the distance between through-hole arrays on the GCPW bandwidth are studied in detail. To make the GCPW more stable and accurate in measurement and reduce the risk of plate bending and deformation, a support layer structure is added to increase the GCPW plate thickness. A GCPW model is constructed using COMSOL simulation software based on the analysis of electromagnetic field theory, and the parameters of the through-hole array are adjusted to improve the bandwidth. Next, a GCPW is fabricated based on the simulation data, and its S-parameters are measured using a vector network analyzer. The measurement results show that the bandwidth of the GCPW developed in this paper is DC to 60 GHz, the insertion loss (S21) > −3 dB, and the return loss (S11) < −10 dB.

## 2. Mechanism of How GCPW’s Through-Hole Structure Spreads Its Bandwidth

Figure 1 shows the structure of GCPW without a through-hole, which causes the excitation of higher-order modes. The loss of higher-order modes and radiation loss causes the GCPW to experience energy leakage [9,10,11,12]. The higher the frequency (greater than or equal to several tens of GHz), the more serious the leakage, thus making it difficult to widen the bandwidth. The introduction of a through-hole array structure in the GCPW can reduce the higher-order modes [11], which makes the resonant frequency move to higher frequencies, reduce energy leakage, and facilitate impedance matching, thus spreading the bandwidth. In this paper, the parameters of the through-hole array [11,13] are changed along the central conductor of the GCPW to suppress the higher-order modes and radiation, reduce the energy leakage, and thus widen the GCPW bandwidth.

Figure 2 shows a schematic diagram of the GCPW with a through-hole array. The parameter *S* indicates the width of the central signal conductor, *G* indicates the width of the slot line, *h* indicates the thickness of the dielectric, *t* is the thickness of the copper foil, the parameter *VR* indicates the radius of the through-hole, and *VD* indicates the through-hole to through-hole distance. The parameter *VL* is the distance from the slotted line to the through-hole array. The through-hole array consists of a row of through-holes located on a specific side ground plane. The parameters *VE1* and *VE2* indicate the distances between the edges of the GCPW ends to their nearest through-holes.

According to the electromagnetic wave propagation theory, the wavelength of electromagnetic wave propagation in the medium is solved by
(1)$$\lambda_{M}=\frac{v_{M}}{f}\approx \frac{c}{\sqrt{\varepsilon_{r}}f}$$
where $\lambda_{M}$ is the wavelength of electromagnetic waves propagating in the medium; $v_{M}$ is the speed of propagation of electromagnetic waves in the medium; f is the frequency of electromagnetic waves; c is the speed of light; and $\varepsilon_{r}$ is the substrate dielectric constant.

In this paper, the design simulation bandwidth is 67 GHz GCPW, and the 67-GHz electromagnetic wave in the medium has a propagation wavelength of about 2.4 mm, as seen in Equation (1). Considering that the length of the patch antenna is generally a half wavelength or an integer times the wavelength when the radiation is at its maximum [14], to determine the length of GCPW, we can consider the integer multiple of the wavelength first followed by the actual demand to suppress radiation. For an electromagnetic wave of 67 GHz, the wavelength propagated in the medium is about 2.4 mm, so 24 mm can be selected as the length of the GCPW when choosing the integer multiple of the wavelength.

Based on [2], the dielectric material of GCPW is determined to be RO4003C, the dielectric constant ($\varepsilon_{r}$) of RO4003C substrate is 3.38, the thickness of the substrate (*h*) is 8 mil, the width of the center conductor (*S*) is 0.35 mm, the width of slot line (*G*) is 0.2 mm, the width (*W*) is 500 mil, and the length *(L*) is 24 mm. Figure 3 shows the simulated S21 and S11 curves for the GCPW without the through-hole. The bandwidth of the GCPW without through-holes is 12.25 GHz, which is far from the design target. The GCPW simulation bandwidth index is the highest when S21 > −3 dB and S11 < −20 dB.

## 3. Research and Design of the Through-Hole Structure to Widen the GCPW Bandwidth

In order to improve the signal transmission efficiency and extend the bandwidth, it is necessary to determine the dielectric substrate material, reduce the conductor loss and dielectric loss, as well as ensure the impedance matching of the characteristic impedance of the through-hole-free GCPW. However, the through-hole-free GCPW structure causes excitation of higher-order modes and energy leakage, making it difficult to extend the bandwidth no matter how the dielectric material and basic structure parameters are changed. To solve this problem, a through-hole array structure can be introduced to form a GCPW with a through-hole structure, so that the equivalent circuit parameters of the GCPW can be changed and the characteristic impedance can be adjusted, thus spreading the bandwidth of the GCPW [15]. The presence of vias can change the equivalent circuit parameters of the GCPW by adding inductance and capacitance, thus changing the characteristic impedance and electromagnetic field distribution of the GCPW. By adding a single-row through-hole array, the resonant frequency can be shifted backward, and impedance matching can be achieved by adjusting the number and position of through-holes. Therefore, the introduction of the through-hole array structure on the non-through-hole GCPW is an effective method to solve the problems of high-order mode excitation and energy leakage and to widen the bandwidth.

The empirical formula for the maximum distance *VD*_max_ between two through-holes is as follows
(2)$$VD_{\max}=\frac{c_{0}}{2f_{\max}\sqrt{\varepsilon_{r}}}$$
where c is the speed of light; $f_{\max}$ is the maximum frequency; and $\varepsilon_{r}$ is the dielectric constant of the substrate material. The bandwidth of the GCPW studied in the simulation of this paper is 67 GHz, and the maximum spacing between through-holes (*VD*_max_) is calculated to be 1.2 mm using Equation (2). Based on the above discussion, the simulation is chosen in the range of *VD* ≤ 1 mm.

### 3.1. Effect of Distance between Through-Hole and Slot Line

GCPW can be made to suppress higher-order modes and radiation by adding through-holes placed along the center conductor and varying the distance from the through-holes to the slot line to spread the bandwidth [16].

When multiple through-holes are placed along the center conductor, an electrical wall and a rectangular waveguide are formed at the GCPW. The cutoff frequency of the main rectangular waveguide is shown in Equation (3) [16,17]:(3)$$f_{c}=\frac{c}{2\sqrt{\varepsilon_{r}}(d_{x}-d)}$$
where $f_{c}$ is the cutoff frequency of the dominant mode; $d_{x}$ is the distance between the two through-holes located opposite the central conductor; d is the through-hole diameter; c is the speed of light; and $\varepsilon_{r}$ is the relative permittivity of the substrate. From [16], it is known that the higher the frequency, the greater the corresponding higher-order mode and radiation at the same cutoff frequency. Therefore, the higher cutoff frequency of the rectangular waveguide suppresses the radiation loss and the propagation of higher-order modes. The bandwidth can be spread by reducing the distance from the through-hole to the slot line.

The *VL* in this section varies from 5.2 mm to 0.2 mm, where *VD* = 1 mm, *VE1 = VE2* = 2.1 mm, and *VR* = 0.3 mm. Figure 4 shows S21 and S11 of the GCPW for different *VL* values and the GCPW bandwidth corresponding to different *VL* values. The smaller the *VL*, the higher the GCPW bandwidth. However, due to the actual machining process, *VL* cannot be infinitely small, so in this paper, after simulating different values of *VL*, the optimal value of *VL* is determined to be 0.2 mm.

### 3.2. Effect of Through-Hole-to-Through-Hole Distance

The through-hole array connects the upper and lower ground plane because the current generates a magnetic field, and there is a parasitic inductance that causes a potential difference that leads to radiation; the potential difference between the upper and lower ground plane causes the propagation of higher-order modes due to the parasitic inductance of the through-hole [16]. The number of through-holes [13] can balance the potential difference between the upper and lower ground plane, reduce the propagation of higher-order modes, change the resonant frequency, and reduce energy leakage. Given that a dense through-hole along the center conductor can reduce radiation loss and achieve a sufficiently good grounding [18], it is beneficial to widen the bandwidth by increasing the number of through-holes and reducing *VD*.

In this section, *VD* is varied from 1 mm to 0.46 mm, keeping *VL* = 0.2 mm, *VE1 = VE2* = 2.1 mm, and *VR* = 0.3 mm. Figure 5 shows S21 and S11 of GCPW with different *VD* values and the bandwidth of GCPW corresponding to different *VD* values. The bandwidth of GCPW increases when *VD* decreases. However, a very small *VD* causes a large impedance mismatch and leads to a reduced bandwidth. In this section, by simulating different *VD* values, the best value of *VD* is confirmed to be 0.6 mm.

### 3.3. Effect of the Distance (VE) between the Ends of the GCPW and the Edge of the Through-Hole Array

In this section, *VE1* varies from 2.1 mm to 0.208 mm and *VE2* varies from 2.1 mm to 0.692 mm. Figure 6 shows the electric field intensity distribution for *VE1 = VE2* = 2.1 mm and *VE1* = 0.208 mm and *VE2* = 0.692 mm. Significant energy leakage occurs at *VE1* = 2.1 mm, so the performance of the GCPW worsens. Therefore, the GCPW bandwidth can be enhanced by decreasing *VE*.

Figure 7 shows the GCPW S21 and S11 plots for different *VE1* and *VE2* values, demonstrating that it is possible to spread the GCPW bandwidth by changing the values of *VE1* and *VE2*.

From the above theoretical analysis and simulation results, this paper finally adopts the parameters of *VL* = 0.2 mm, *VD* = 0.6 mm, *VE1* = 0.208 mm, and *VE2* = 0.692 mm to process and fabricate the single-row through-hole array of the GCPW. The simulation bandwidth of the GCPW reaches 67 GHz, as shown in Figure 8. Figure 9 shows the physical diagram of the single-row GCPW. The simulation bandwidth of GCPW without through-hole is 12.25 GHz, and the simulation bandwidth of GCPW with through-hole array is 67 GHz, so the widening ratio of array with through-hole is 446.93%.

## 4. Improving the GCPW’s Performance by Increasing the Number of Through-Hole Arrays

Due to the high through-hole density of the single-row through-hole array, the thermal expansion coefficient of the PCB board may not match, causing the PCB board to be deformed, lead to through-hole fracture, and adversely affect the mechanical properties of the PCB board. To ensure that the transmission performance of the GCPW is unchanged and easy to process, the multi-row through-hole array of GCPW is proposed [19]. Moreover, the through-hole data of the multi-row through-hole array structure are not as dense as the data of the single-row array structure, so the mechanical strength and processing conditions are more favorable than those of the single-row through-hole array. Therefore, this paper proposes a multi-row through-hole array to improve the mechanical properties of the PCB while ensuring the GCPW is easy to process and refraining from degrading the performance of the GCPW.

As shown in Figure 10, additional through-hole arrays are added to the existing structure to improve the GCPW performance. The spacing (*RD*) of the through-hole arrays can be varied. The first through-hole array is closest to the center signal line, while the third through-hole array is furthest from the center signal line. The second through-hole array is offset VX from the first through-hole array along the x-axis, and the *VX* value is half of *VD*, with the distance between the through-holes being *VD*.

### 4.1. Effect of Different Through-Hole Rows on the Bandwidth

Figure 11 shows that within a certain range, the greater the number of rows in the through-hole array, the higher the bandwidth. However, when there are too many rows, the effect of enhancing the bandwidth is not obvious, so it is necessary to find the appropriate number of rows to enhance the bandwidth, so in this paper, two and three rows of through-hole arrays of GCPWs are selected for processing.

### 4.2. Effect of Distance (RD) between Through-Hole Arrays

From Figure 12, it is concluded that the smaller the distance between the through-hole arrays within a certain range, the higher the GCPW bandwidth. However, note that if the through-hole arrays are too close to each other, the problem of impedance mismatch occurs, so the reflection increases and the bandwidth decreases. Therefore, when determining the *RD* distance, care must be taken to balance both impedance matching and bandwidth enhancement to make the bandwidth 67 GHz.

### 4.3. Main Parameters of Double-Row and Triple-Row Through-Hole Array GCPWs with the Bandwidth of 67 GHz

The basic parameters of the GCPWs with double-row and triple-row through-hole arrays with the simulated bandwidth of 67 GHz proposed in this paper are the same as those of the single-row GCPW. Other parameters are *VE1 = VE2* = 0.3 mm, *VD* = 0.692 mm, *VL* = 0.246 mm, *VR* = 10 mil, and *RD* = 0.854 mm. The process of determining other parameters: first add a single-row through-hole array, determine the number of through-holes and the radius of the single-row through-hole array according to the parameters of the 50 GHz GCPW proposed by Jiangmiao Zhu [2], then extend the GCPW bandwidth by adjusting the *VL*, *VD* and *VE* parameters, and then extend the bandwidth to 67 GHz by increasing the number of through-hole arrays and adjusting the *RD* parameters. The parameters of single-row through-hole array GCPW are denser than those of the double-row and triple-row through-hole array GCPWs. Therefore, the double-row and triple-row GCPWs are easier to process. Figure 13 shows the physical diagram of the machining of the double-row and triple-row GCPWs.

## 5. Improving the GCPW’s Performance by Adding Support Layers

A support layer is added to the GCPW to stabilize its structure and make it less prone to deformation [19], thereby enabling it to make more accurate measurements. The support layer consists of two parts of material, divided into two layers: the first part of the material is FSD330N with thickness *T* and the second part of the material is RO4003C, a high-frequency material with thickness *H*. Figure 14 presents the cross-sectional view of the GCPW with the support layer added. However, because the signal is transmitted over the GCPW, the propagation speed changes due to the change of the medium reflected by the signal and the change of the equivalent dielectric constant, causing the GCPW bandwidth to change.

Therefore, several simulation experiments are conducted to prove the effect of the thickness of the transforming material on the bandwidth. First, the original thickness of the support layer is determined. Because the thickness of the GCPW medium is 8 mil, the variables in the first simulation are set to *T = H* = 0.5 mm. Next, *H* is kept constant, and the value of *T* is varied. As can be seen from Figure 15, when *T* is smaller, the bandwidth is higher, so it is desirable to reduce the value of *T* to 0.1 mm. Figure 16 shows that reducing the thickness of *H* has little effect on the bandwidth, so after considering the cost and processing conditions, the most desirable value of *H* is determined to be 0.2 mm. At this time, the simulation bandwidth of the GCPW is 63.5 GHz. Although adding the support layer decreases the bandwidth of the GCPW, it increases the stability of the GCPW, so adding the support layer is necessary.

## 6. S-Parameter Measurement of GCPW

### 6.1. Measurement Process of the Vector Network Analyzer

After processing a single-row GCPW without a support layer, the S21 and S11 of GCPW are measured with a keysight E8361A vector network analyzer. The temperature during the measurement was 24 degrees and the humidity was 45% at the metrology institute. The vector network analyzer is first calibrated, and the measurement is carried out after the calibration is finished. The measurement interval is set to 50 MHz, the starting frequency is 50 MHz, and the cutoff frequency is 67 GHz. The connectors at both ends of the GCPW are connected to the coaxial cable port for measurement, and the curves of S11 and S21 are obtained and stored. The measurement diagram of the vector network analyzer is shown in Figure 17.

### 6.2. Comparison of Measurement Results and Simulation Results

As can be seen from Figure 18, Figure 19 and Figure 20, the actual measurements in the frequency range from DC to 60 GHz are satisfied with S21 > −3 dB and S11 > −10 dB, indicating that the GCPW measured in this paper has good performance in this frequency range with a bandwidth of 60 GHz. By comparing the results in these three figures, it can be found that a similar fluctuation change is observed at the frequency position after 30 GHz, indicating that this variation may be due to only one reason, and therefore, it is inferred that this variation may be caused by the performance problem of the connector.

In addition, the simulated bandwidth of the GCPWs is 67 GHz, which is different from the measured bandwidth of the vector network at 60 GHz. This may be caused by the slight difference between the connector performance, the manufactured and simulated values of the processing, and the effect of the copper foil roughness on the signal transmission.

In the frequency range from 60 GHz to 67 GHz, the S21 of GCPW No.1, No.2 and No.3 is less than −3 dB, but the maximum “bad point” is −3.83 dB, and only a very few points of S11 is greater than −10 dB, so although there is a certain signal loss, but because the loss is small, it is acceptable. Therefore, although there is a certain amount of signal loss, but because the loss is small, it is acceptable, and these GCPW can still be used in actual engineering.

From the frequency range of DC-67 GHz, the maximum “bad point” of S21 of the three GCPWs is −3.83 dB, and because this loss is small, it does not affect the use in the frequency range of DC-67 GHz. In the actual measurement, the calibration requirement has been satisfied for the probe calibration experiment with the bandwidth below 20 GHz, and the design of single-row, two-row and three-row unsupported layer GCPWs in this paper is successful. The GCPWs, which have been measured in the frequency domain, can be used to calibrate digital oscilloscope probes. Therefore, the GCPWs measured in this paper can be used to calibrate the probes of digital oscilloscopes. Calibration of oscilloscope probes can be accomplished by building a broadband GCPW-based measurement system, obtaining data through experiments, and then obtaining the bandwidth of the probe through data processing.

## 7. Conclusions

In this paper, broadband GCPWs with different types of through-hole arrays were designed and fabricated. Furthermore, the effect of changing the geometry of the through-holes on the GCPW bandwidth was investigated through a theoretical analysis of electromagnetic fields and COMSOL simulation; the main parameters for spreading the GCPW bandwidth were also analyzed in detail. In addition, when the main parameters of the wideband GCPW were too small to be fabricated due to the limitation of the fabrication process, increasing the number of rows of through-hole arrays was proposed to widen the frequency band. Three physical GCPWs were fabricated according to the simulation data, which were measured with a vector network analyzer, and bandwidths ranging from DC to 60 GHz were obtained, all of which satisfied S21 > −3 dB and S11 > −10 dB, thus meeting the design requirements.

## Figures and Tables

**Figure 1 sensors-23-04342-f001:**
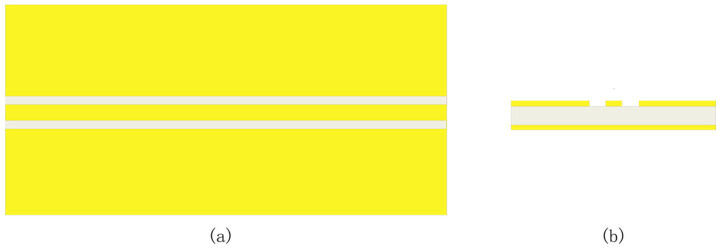
A grounded coplanar waveguide (GCPW) without through-hole. (**a**) Top view. (**b**) Side view.

**Figure 2 sensors-23-04342-f002:**
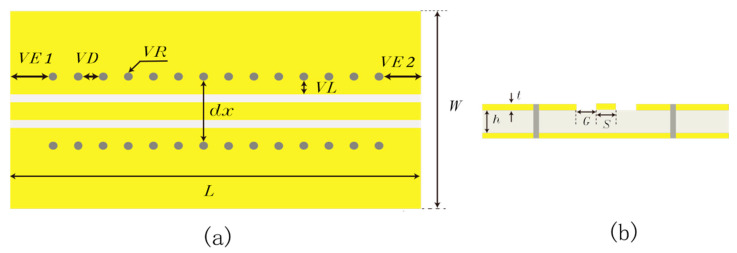
GCPW with the through-hole array. (**a**) Top view. (**b**) Side view.

**Figure 3 sensors-23-04342-f003:**
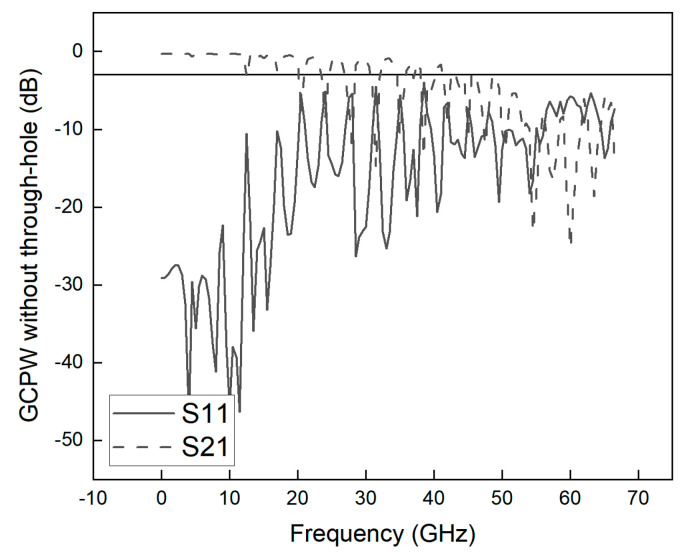
Parameters S21 and S11 for GCPW without through-hole.

**Figure 4 sensors-23-04342-f004:**
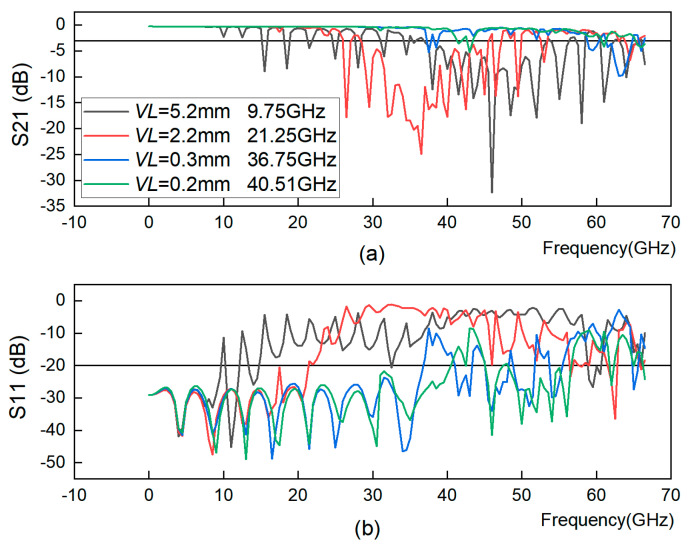
GCPW under different through-hole-to-slot line distance (*VL*) conditions. (**a**) S21 simulation diagram. (**b**) S11 simulation diagram.

**Figure 5 sensors-23-04342-f005:**
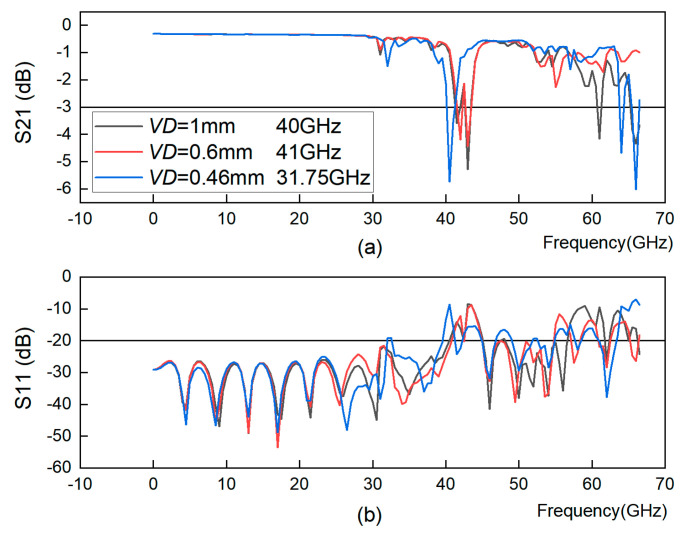
GCPW with different through-hole to through-hole distances *VD*. (**a**) S21 simulation; (**b**) S11 simulation.

**Figure 6 sensors-23-04342-f006:**
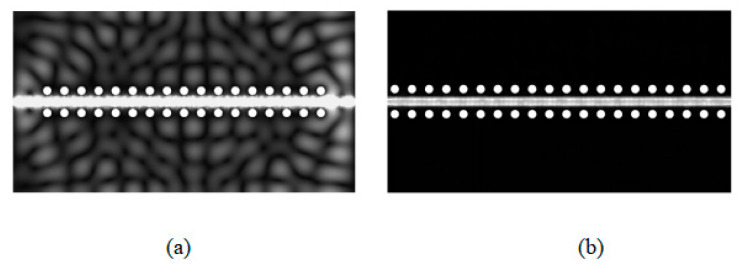
GCPW electric field diagram at 66.5 GHz for (**a**) *VE1* = *VE2* = 2.1 mm and (**b**) *VE1* = 0.208 mm and *VE2* = 0.692 mm.

**Figure 7 sensors-23-04342-f007:**
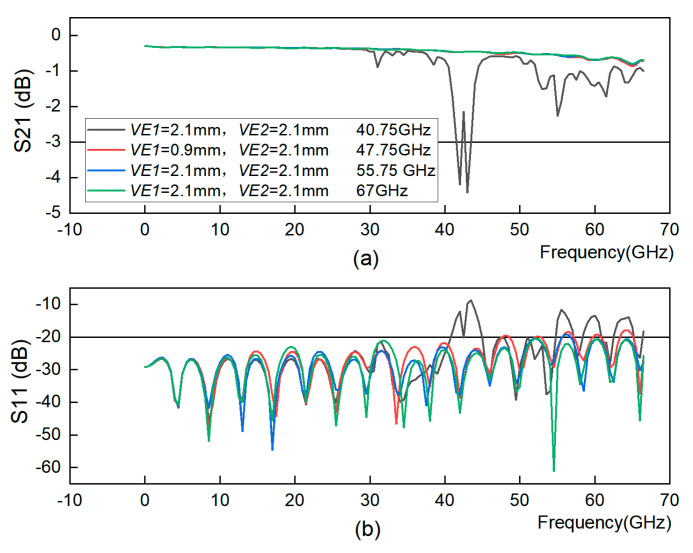
The distance between the edges of the two ends of the GCPW and its nearest through-hole. (**a**) S21 simulation. (**b**) S11 simulation.

**Figure 8 sensors-23-04342-f008:**
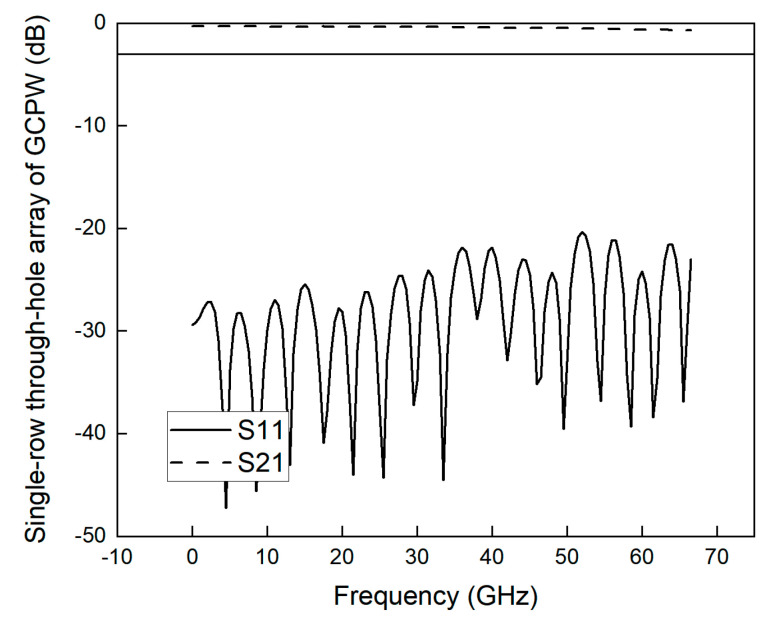
S21 simulation and S11 simulation of single-row GCPW.

**Figure 9 sensors-23-04342-f009:**
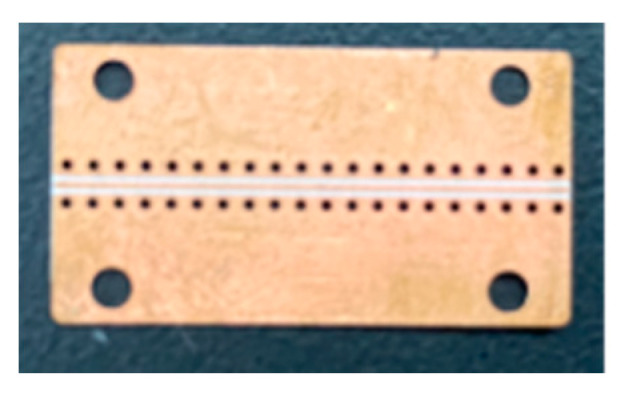
Physical diagram of single-row through-hole array of GCPW.

**Figure 10 sensors-23-04342-f010:**
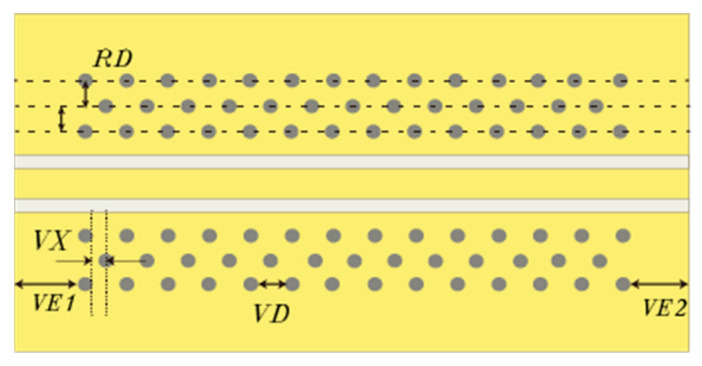
Top view of GCPW with different through-hole array structures.

**Figure 11 sensors-23-04342-f011:**
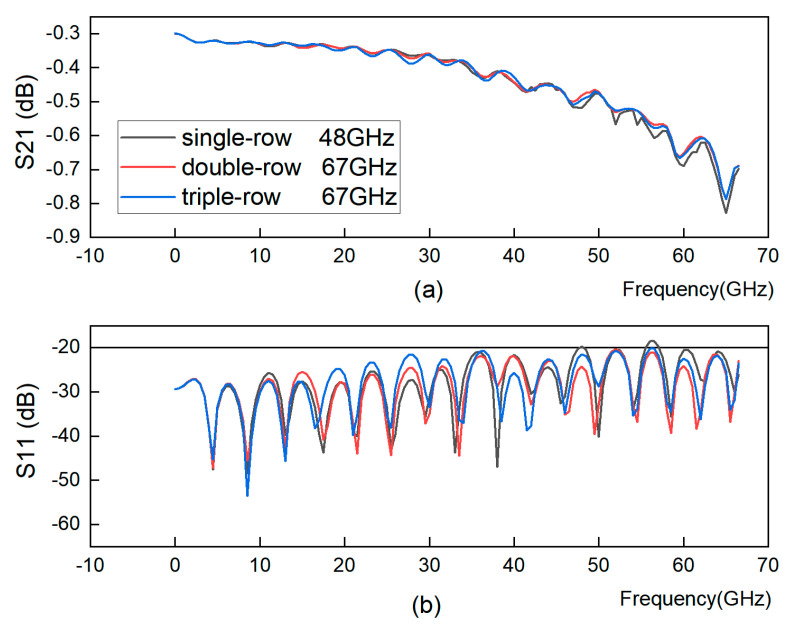
GCPW with different rows of through-hole arrays. (**a**) S21 simulation. (**b**) S11 simulation.

**Figure 12 sensors-23-04342-f012:**
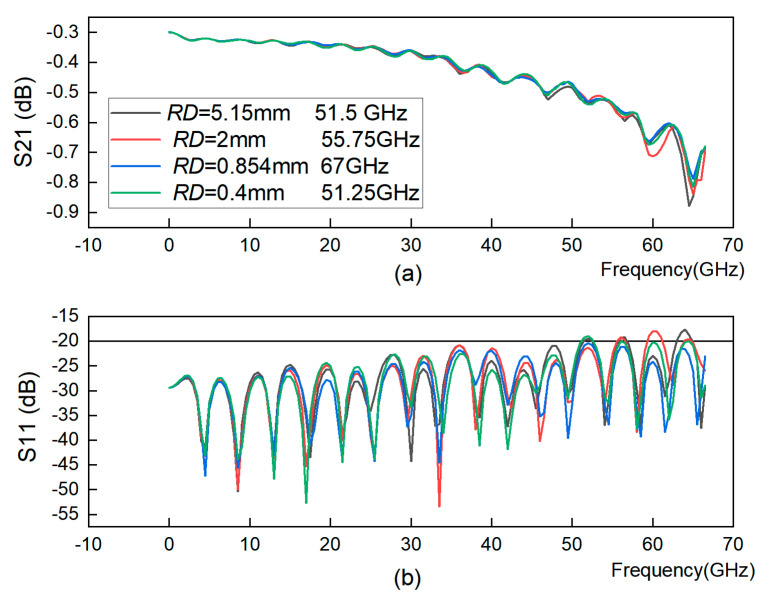
GCPW with double-row through-hole and varying inter-row distance. (**a**) S21 simulation. (**b**) S11 simulation.

**Figure 13 sensors-23-04342-f013:**
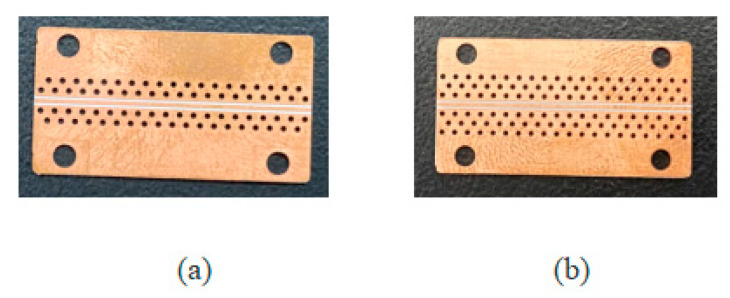
Physical view of GCPWs with (**a**) the double-row through-hole array and (**b**) the triple-row through-hole array.

**Figure 14 sensors-23-04342-f014:**
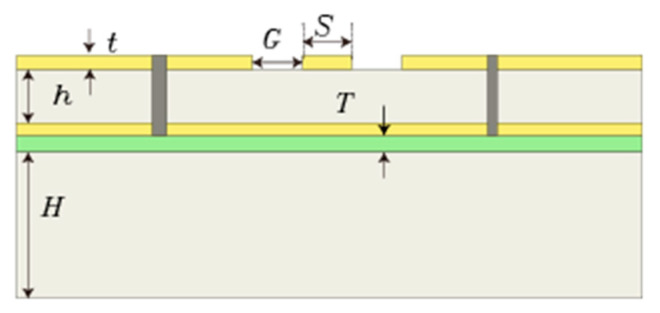
Cross-sectional image of GCPW with the support layer.

**Figure 15 sensors-23-04342-f015:**
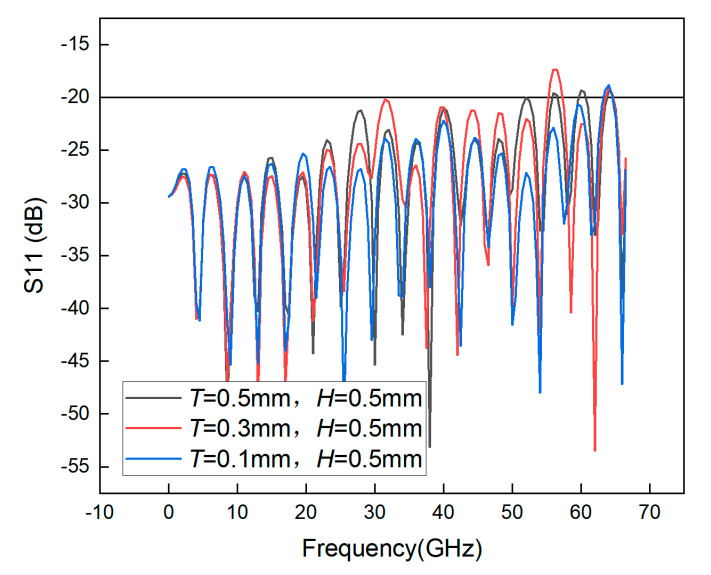
S11 simulation of GCPW with different thicknesses *T*.

**Figure 16 sensors-23-04342-f016:**
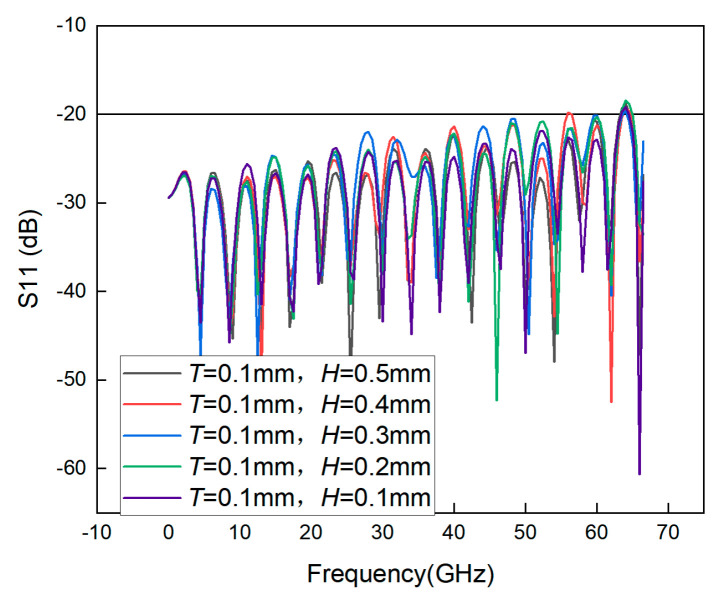
S11 simulation of GCPW with different thicknesses *H*.

**Figure 17 sensors-23-04342-f017:**
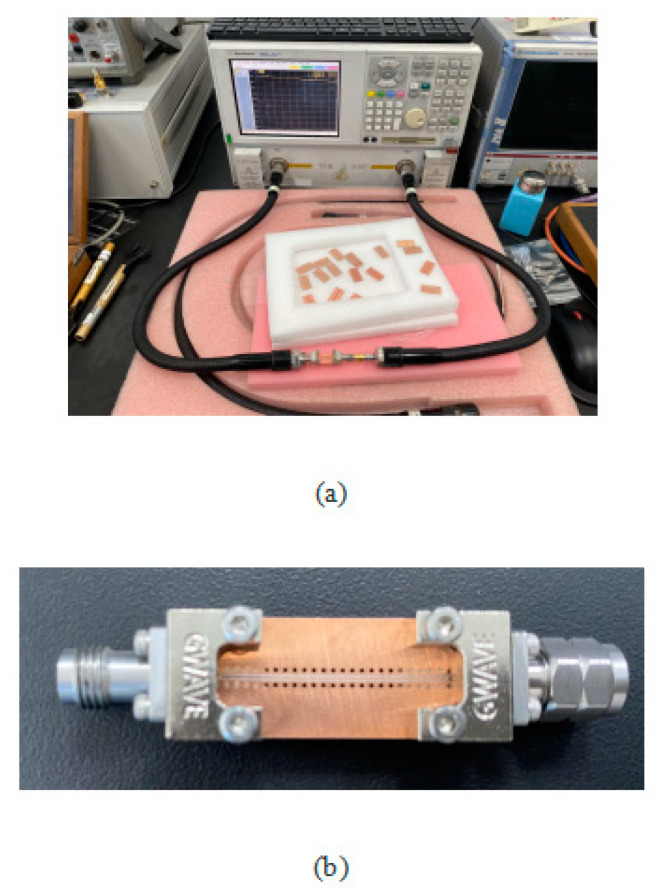
(**a**) Measured vector network analyzer plot; (**b**) Measured GCPW plot.

**Figure 18 sensors-23-04342-f018:**
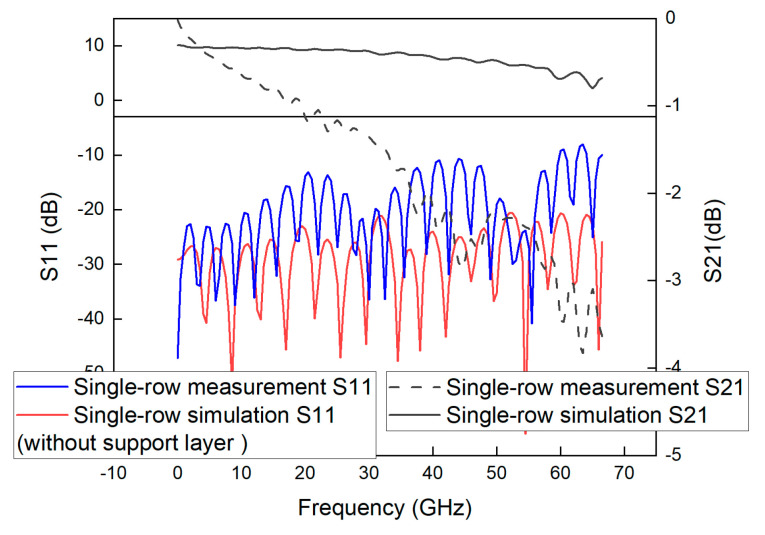
Simulated and measured S-parameters of GCPW with a single-row unsupported layer through-hole array.

**Figure 19 sensors-23-04342-f019:**
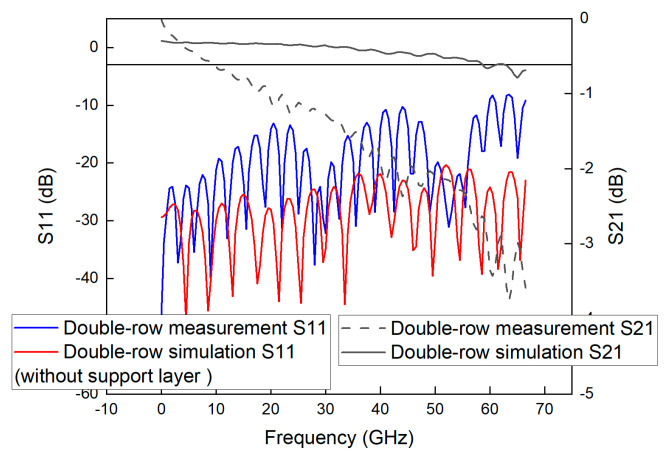
Simulated and measured S-parameters of GCPW with a double-row unsupported layer through-hole array.

**Figure 20 sensors-23-04342-f020:**
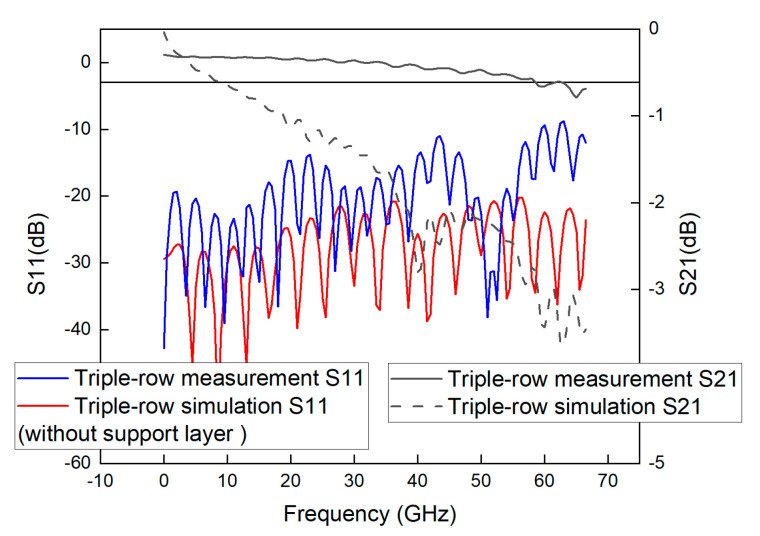
Simulated and measured S-parameters of GCPW with a triple-row unsupported layer through-hole array.

## Data Availability

Data underlying the results presented in this paper are not publicly available at this time but may be obtained from the authors upon reasonable request.

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
