# Peer review of "Method to Change the Through-Hole Structure to Broaden Grounded Coplanar Waveguide Bandwidth"

_sensors, 2023, doi:10.3390/s23094342_

Round 1

Reviewer 1 Report

In overall presentation of the manuscript must be improve, including text and formatting. Proposed of through-hole to broaden CPW bandwidth as presented need clearly step-by-step elaborate, how the proposed method effect to the bandwidth (every single hole) then percentage broaden as well as impact to the antenna performance. Figure 18,19 and 20 shows the original results before introduce through-hole has ready broad bandwidth and need to justify how much target of bandwidth as well as the application.

Reviewer 2 Report

The article "Method to change the through-hole structure to broaden grounded coplanar waveguide bandwidth" proposes a method to change the through-hole structure to widen grounded coplanar waveguide to be used to calibrate broadband oscilloscope probes. In this reviewer’s opinion, the work needs improvements:

1) Figure 3 shows simulated or measured curves?

2) What are room conditions during the measurements (temperature and humidity)?

3) In the Introduction, I recommend inserting a brief discussion and examples of state-of-art microwave sensors applications using microwave and sensors integrated with other technique. This is a good direction for future works and can be justify the proposed calibration method. Suggestions: 10.1109/JSEN.2022.3202708 ; 10.1109/SENSORS52175.2022.9966998

4) Insert space in subtitles text Figure18, Figure19 and Figure20 => Figure 18, Figure 19 and Figure 20

5) The article not includes details about the oscilloscope calibration. Insert the details about the oscilloscope calibration.

Reviewer 3 Report

In this manuscript, the authors investigated the method to improve the bandwidth of GCPW by changing the through-hole structure. The effect of adjusting parameters of through-hole arrays is studied through simulations. GCPWs with various through-hole structures were fabricated, measured, and exhibited satisfactory performance based on the measured results. Revision is required before publication. A list of specific comments is as follows: 

1. In the figures showing the top view and side view of GCPW, the color code used for copper and substrate should match.

2. It is hard to read the multiple curves plotted by dashed, dotted lines, e.g. Figure 4. Please consider using colors or other distinguishable methods to plot multiple curves in one figure. 

3. Figure 7 is a duplication of Figure 6

4. How were the parameters in lines 229 and 230 other than RD determined?

5. When showing S21 and S11 in one figure, consider using the left y-axis for one parameter and the right y-axis for the other with the different ranges. So it will be clearer to see the variations of S21, especially in Figure 18-20.

Round 2

Reviewer 1 Report

No more comment and recommend to publish

Author Response

Thank you very much for your reply, for reading the paper and taking up your time, and I wish you all the best.

Reviewer 3 Report

Thank you for the revision and responses. The authors have addressed most of my concerns, but I still have a few concerns about the figures.

1. Figures 10 and 14 should have the same color style as Figure 2.

2. Figures 18-20 need to be double-checked. Figure 18 in the manuscript and the response letter look different. The horizontal lines need to be moved after changing the y-axis range. There are two black solid lines and two black dashed lines in one figure, S21 and S11 curves need to be shown in different line styles or colors.
